# Neural Sparse Voxel Fields

**Lingjie Liu**[†*]**, Jiatao Gu**[‡*]**, Kyaw Zaw Lin**[◇]**, Tat-Seng Chua**[◇]**, Christian Theobalt**[†]
[†]Max Planck Institute for Informatics
[‡]Facebook AI Research   [◇]National University of Singapore
[†]{lliu,theobalt}@mpi-inf.mpg.de
[‡]jgu@fb.com   [◇]kyawzl@comp.nus.edu.sg
[◇]dcscts@nus.edu.sg

## Abstract

Photo-realistic free-viewpoint rendering of real-world scenes using classical computer graphics techniques is challenging, because it requires the difficult step of capturing detailed appearance and geometry models. Recent studies have demonstrated promising results by learning scene representations that implicitly encode both geometry and appearance without 3D supervision. However, existing approaches in practice often show blurry renderings caused by the limited network capacity or the difficulty in finding accurate intersections of camera rays with the scene geometry. Synthesizing high-resolution imagery from these representations often requires time-consuming optical ray marching. In this work, we introduce *Neural Sparse Voxel Fields (NSVF)*, a new neural scene representation for fast and high-quality free-viewpoint rendering. NSVF defines a set of voxel-bounded implicit fields organized in a sparse voxel octree to model local properties in each cell. We progressively learn the underlying voxel structures with a diffentiable ray-marching operation from only a set of posed RGB images. With the sparse voxel octree structure, rendering novel views can be accelerated by skipping the voxels containing no relevant scene content. Our method is typically over 10 times faster than the state-of-the-art (namely, NeRF (Mildenhall et al., 2020)) at inference time while achieving higher quality results. Furthermore, by utilizing an explicit sparse voxel representation, our method can easily be applied to scene editing and scene composition. We also demonstrate several challenging tasks, including multi-scene learning, free-viewpoint rendering of a moving human, and large-scale scene rendering.

## 1   Introduction

Realistic rendering in computer graphics has a wide range of applications including mixed reality, visual effects, visualization, and even training data generation in computer vision and robot navigation. Photo-realistically rendering a real world scene from arbitrary viewpoints is a tremendous challenge, because it is often infeasible to acquire high-quality scene geometry and material models, as done in high-budget visual effects productions. Researchers therefore have developed image-based rendering (IBR) approaches that combine vision-based scene geometry modeling with image-based view interpolation (Shum and Kang, 2000; Zhang and Chen, 2004; Szeliski, 2010). Despite their significant progress, IBR approaches still have sub-optimal rendering quality and limited control over the results, and are often scene-type specific. To overcome these limitations, recent works have employed deep neural networks to implicitly learn scene representations encapsulating both geometry and appearance from 2D observations with or without a coarse geometry. Such neural representations are commonly combined with 3D geometric models, such as voxel grids (Yan et al., 2016; Sitzmann et al., 2019a;

---

[*]Equal contribution.

Lombardi et al., 2019), textured meshes (Thies et al., 2019; Kim et al., 2018; Liu et al., 2019a, 2020), multi-plane images (Zhou et al., 2018; Flynn et al., 2019; Mildenhall et al., 2019), point clouds (Meshry et al., 2019; Aliev et al., 2019), and implicit functions (Sitzmann et al., 2019b; Mildenhall et al., 2020).

Unlike most explicit geometric representations, neural implicit functions are smooth, continuous, and can - in theory - achieve high spatial resolution. However, existing approaches in practice often show blurry renderings caused by the limited network capacity or the difficulty in finding accurate intersections of camera rays with the scene geometry. Synthesizing high-resolution imagery from these representations often requires time-consuming optical ray marching. Furthermore, editing or re-compositing 3D scene models with these neural representations is not straightforward.

In this paper, we propose *Neural Sparse Voxel Fields* (NSVF), a new implicit representation for fast and high-quality free-viewpoint rendering. Instead of modeling the entire space with a single implicit function, NSVF consists of a set of voxel-bounded implicit fields organized in a sparse voxel octree. Specifically, we assign a voxel embedding at each vertex of the voxel, and obtain the representation of a query point inside the voxel by aggregating the voxel embeddings at the eight vertices of the corresponding voxel. This is further passed through a multilayer perceptron network (MLP) to predict geometry and appearance of that query point. Our method can progressively learn NSVF from coarse to fine with a differentiable ray-marching operation from only a set of posed 2D images of a scene. During training, the sparse voxels containing no scene information will be pruned to allow the network to focus on the implicit functions learning for volume regions with scene contents. With the sparse voxels, rendering at inference time can be greatly accelerated by skipping empty voxels without scene content.

Our method is typically over 10 times faster than the state-of-the-art (namely, NeRF (Mildenhall et al., 2020)) at inference time while achieving higher quality results. We extensively evaluate our method on a variety of challenging tasks including multi-object learning, free-viewpoint rendering of dynamic and indoor scenes. Our method can be used to edit and composite scenes. To summarize, our technical contributions are:

- We present NSVF that consists of a set of voxel-bounded implicit fields, where for each voxel, voxel embeddings are learned to encode local properties for high-quality rendering;

- NSVF utilizes the sparse voxel structure to achieve efficient rendering;

- We introduce a progressive training strategy that efficiently learns the underlying sparse voxel structure with a differentiable ray-marching operation from a set of posed 2D images in an end-to-end manner.

## 2 Background

Existing neural scene representations and neural rendering methods commonly aim to learn a function that maps a spatial location to a feature representation that implicitly describes the local geometry and appearance of the scene, where novel views of that scene can be synthesized using rendering techniques in computer graphics. To this end, the rendering process is formulated in a differentiable way so that the neural network encoding the scene representation can be trained by minimizing the difference between the renderings and 2D images of the scene. In this section, we describe existing approaches to representation and rendering using implicit fields and their limitations.

### 2.1 Neural Rendering with Implicit Fields

Let us represent a scene as an implicit function $F_\theta: (\boldsymbol{p}, \boldsymbol{v}) \rightarrow (\boldsymbol{c}, \omega)$, where $\theta$ are parameters of an underlying neural network. This function describes the scene color $\boldsymbol{c}$ and its probability density $\omega$ at spatial location $\boldsymbol{p}$ and ray direction $\boldsymbol{v}$. Given a pin-hole camera at position $\boldsymbol{p}_0 \in \mathbb{R}^3$, we render a 2D image of size $H \times W$ by shooting rays from the camera to the 3D scene. We thus evaluate a volume rendering integral to compute the color of camera ray $\boldsymbol{p}(z) = \boldsymbol{p}_0 + z \cdot \boldsymbol{v}$ as:

$$\boldsymbol{C}(\boldsymbol{p}_0, \boldsymbol{v}) = \int_0^{+\infty} \omega(\boldsymbol{p}(z)) \cdot \boldsymbol{c}(\boldsymbol{p}(z), \boldsymbol{v}) dz, \quad \text{where} \quad \int_0^{+\infty} \omega(\boldsymbol{p}(z)) dz = 1 \qquad (1)$$

Note that, to encourage the scene representation to be multiview consistent, $\omega$ is restricted as a function of only $\boldsymbol{p}(z)$ while $\boldsymbol{c}$ takes both $\boldsymbol{p}(z)$ and $\boldsymbol{v}$ as inputs to model view-dependent color. Different rendering strategies to evaluate this integral are feasible.

**Surface Rendering.** Surface-based methods (Sitzmann et al., 2019b; Liu et al., 2019b; Niemeyer et al., 2019) assume $\omega(\boldsymbol{p}(z))$ to be the Dirac function $\delta(\boldsymbol{p}(z) - \boldsymbol{p}(z^*))$ where $\boldsymbol{p}(z^*)$ is the intersection of the camera ray with the scene geometry.

**Volume Rendering.** Volume-based methods (Lombardi et al., 2019; Mildenhall et al., 2020) estimate the integral $\boldsymbol{C}(\boldsymbol{p}_0, \boldsymbol{v})$ in Eq. 1 by densely sampling points on each camera ray and accumulating the colors and densities of the sampled points into a 2D image. For example, the state-of-the-art method NeRF (Mildenhall et al., 2020) estimates $\boldsymbol{C}(\boldsymbol{p}_0, \boldsymbol{v})$ as:

$$\boldsymbol{C}(\boldsymbol{p}_0, \boldsymbol{v}) \approx \sum_{i=1}^{N} \left( \prod_{j=1}^{i-1} \alpha(z_j, \Delta_j) \right) \cdot (1 - \alpha(z_i, \Delta_i)) \cdot \boldsymbol{c}(\boldsymbol{p}(z_i), \boldsymbol{v}) \tag{2}$$

where $\alpha(z_i, \Delta_i) = \exp(-\sigma(\boldsymbol{p}(z_i) \cdot \Delta_i))$, and $\Delta_i = z_{i+1} - z_i$. $\{\boldsymbol{c}(\boldsymbol{p}(z_i), \boldsymbol{v})\}_{i=1}^{N}$ and $\{\sigma(\boldsymbol{p}(z_i))\}_{i=1}^{N}$ are the colors and the volume densities of the sampled points.

## 2.2 Limitations of Existing Methods

For surface rendering, it is critically important that an accurate surface is found for learned color to be multi-view consistent, which is hard and detrimental to training convergence so that induces blur in the renderings. Volume rendering methods need to sample a high number of points along the rays for color accumulation to achieve high quality rendering. However, evaluation of each sample points along the ray as NeRF does is inefficient. For instance, it takes around 30 seconds for NeRF to render an $800 \times 800$ image. Our main insight is that it is important to prevent sampling of points in empty space without relevant scene content as much as possible. Although NeRF performs importance sampling along the ray, due to allocating fixed computational budget for every ray, it cannot exploit this opportunity to improve rendering speed. We are inspired by classical computer graphics techniques such as the bounding volume hierarchy (BVH, Rubin and Whitted, 1980) and the sparse voxel octree (SVO, Laine and Karras, 2010) which are designed to model the scene in a sparse hierarchical structure for ray tracing acceleration. In this encoding, local properties of a spatial location only depend on a local neighborhood of the leaf node that the spatial location belongs to. In this paper we show how hierarchical sparse volume representations can be used in a neural network-encoded implicit field of a 3D scene to enable detailed encoding, and efficient, high quality differentiable volumetric rendering, even of large scale scenes.

# 3 Neural Sparse Voxel Fields

In this section, we introduce *Neural Sparse-Voxel Fields* (NSVF), a hybrid scene representation that combines neural implicit fields with an explicit sparse voxel structure. Instead of representing the entire scene as a single implicit field, NSVF consists of a set of voxel-bounded implicit fields organized in a sparse voxel octree. In the following, we describe the building block of NSVF - a voxel-bounded implicit field (§ 3.1) - followed by a rendering algorithm for NSVF (§ 3.2), and a progressive learning strategy (§ 3.3).

## 3.1 Voxel-bounded Implicit Fields

We assume that the relevant non-empty parts of a scene are contained within a set of sparse (bounding) voxels $\mathcal{V} = \{V_1 \ldots V_K\}$, and the scene is modeled as a set of voxel-bounded implicit functions: $F_\theta(\boldsymbol{p}, \boldsymbol{v}) = F_\theta^i(\boldsymbol{g}_i(\boldsymbol{p}), \boldsymbol{v})$ if $\boldsymbol{p} \in V_i$. Each $F_\theta^i$ is modeled as a multi-layer perceptron (MLP) with shared parameters $\theta$:

$$F_\theta^i : (\boldsymbol{g}_i(\boldsymbol{p}), \boldsymbol{v}) \rightarrow (\boldsymbol{c}, \sigma), \forall \boldsymbol{p} \in V_i, \tag{3}$$

Here $\boldsymbol{c}$ and $\sigma$ are the color and density of the 3D point $\boldsymbol{p}$, $\boldsymbol{v}$ is ray direction, $g_i(\boldsymbol{p})$ is the representation at $\boldsymbol{p}$ which is defined as:

$$g_i(\boldsymbol{p}) = \zeta \left( \chi \left( \widetilde{g}_i(\boldsymbol{p}_1^*), \ldots, \widetilde{g}_i(\boldsymbol{p}_8^*) \right) \right) \tag{4}$$

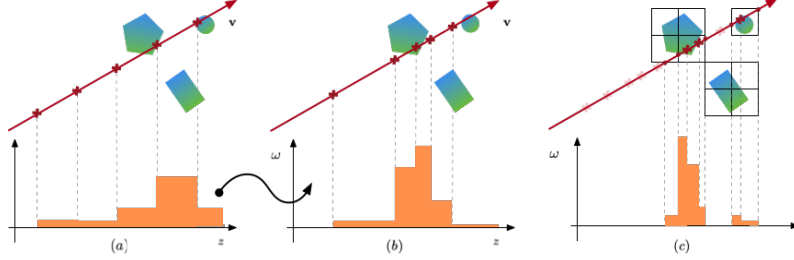

Figure 1: Illustrations of (a) uniform sampling; (b) importance sampling based on the results in (a); (c) the proposed sampling approach based on sparse voxels.

where $\boldsymbol{p}_1^*, \ldots, \boldsymbol{p}_8^* \in \mathbb{R}^3$ are the eight vertices of $V_i$, and $\widetilde{g}_i(\boldsymbol{p}_1^*), \ldots, \widetilde{g}_i(\boldsymbol{p}_8^*) \in \mathbb{R}^d$ are feature vectors stored at each vertex. In addition, $\chi(.)$ refers to trilinear interpolation, and $\zeta(.)$ is a post-processing function. In our experiments, $\zeta(.)$ is positional encoding proposed by (Vaswani et al., 2017; Mildenhall et al., 2020).

Compared to using the 3D coordinate of point $\boldsymbol{p}$ as input to $F_\theta^i$ as most of previous works do, in NSVF, the feature representation $g_i(\boldsymbol{p})$ is aggregated by the eight voxel embeddings of the corresponding voxel where region-specific information (e.g. geometry, materials, colors) can be embedded. It significantly eases the learning of subsequent $F_\theta^i$ as well as facilitates high-quality rendering.

**Special Cases.** NSVF subsumes two classes of earlier works as special cases. (1) When $\widetilde{g}_i(\boldsymbol{p}_k^*) = \boldsymbol{p}_k^*$ and $\zeta(.)$ is the positional encoding, $g_i(\boldsymbol{p}) = \zeta(\chi(\boldsymbol{p}_1^*, \ldots, \boldsymbol{p}_8^*)) = \zeta(\boldsymbol{p})$, which means that NeRF (Mildenhall et al., 2020) is a special case of NSVF. (2) When $\widetilde{g}_i(\boldsymbol{p}) : \boldsymbol{p} \to (\boldsymbol{c}, \sigma)$, $\zeta(.)$ and $F_\theta^i$ are identity functions, our model is equivalent to the models which use explicit voxels to store colors and densities, e.g., Neural Volumes (Lombardi et al., 2019).

### 3.2 Volume Rendering

NSVF encodes the color and density of a scene at any point $\boldsymbol{p} \in \mathcal{V}$. Compared to rendering a neural implicit representation that models the entire space, rendering NSVF is much more efficient as it obviates sampling points in the empty space. Rendering is performed in two steps: (1) ray-voxel intersection; and (2) ray-marching inside voxels. We illustrate the pipeline in Appendix Figure 8,

**Ray-voxel Intersection.** We first apply Axis Aligned Bounding Box intersection test (AABB-test) (Haines, 1989) for each ray. It checks whether a ray intersects with a voxel by comparing the distances from the ray origin to each of the six bounding planes of the voxel. The AABB test is very efficient especially for a hierarchical octree structure (e.g. NSVF), as it can readily process millions of voxels in real time. Our experiments show that $10k \sim 100k$ sparse voxels in the NSVF representation are enough for photo-realistic rendering of complex scenes.

**Ray Marching inside Voxels.** We return the color $\boldsymbol{C}(\boldsymbol{p}_0, \boldsymbol{v})$ by sampling points along a ray using Eq. (2). To handle the case where a ray misses all the objects, we additionally add a background term $A(\boldsymbol{p}_0, \boldsymbol{v}) \cdot \boldsymbol{c}_{\text{bg}}$ on the right side of Eq. (2), where we define *transparency* $A(\boldsymbol{p}_0, \boldsymbol{v}) = \prod_{i=1}^{N} \alpha(z_i, \Delta_i)$, and $\boldsymbol{c}_{\text{bg}}$ is learnable RGB values for background. As discussed in § 2, volume rendering requires dense samples along the ray in non-empty space to achieve high quality rendering. Densely evaluating at uniformly sampled points in the whole space (Figure 1 (a)) is inefficient because empty regions are frequently and unnecessarily tested. To focus on sampling in more important regions, Mildenhall et al. (2020) learned two networks where the second network is trained with samples from the distribution estimated by the first one (Figure 1 (b)). However, this further increases the training and inference complexity. In contrast, NSVF does not employ a secondary sampling stage while achieving better visual quality. As shown in Figure 1 (c), we create a set of query points using rejection sampling based on sparse voxels. Compared to the aforementioned approaches, we are able to sample more densely at the same evaluation cost. We include all voxel intersection points as additional samples and perform color accumulation with the *midpoint rule*. Our approach is summarized in Algorithm 1 where we additionally return the transparency $A$, and the expected depth $Z$ which can be further used for visualizing the normal with finite difference.

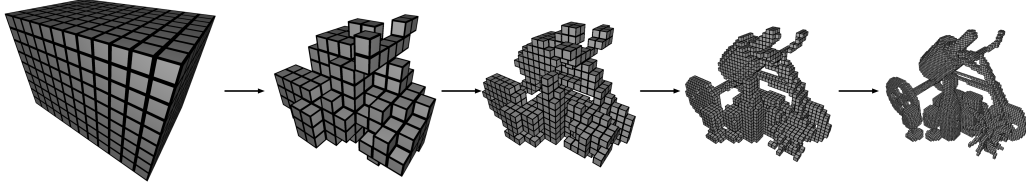

Figure 2: Illustration of self-pruning and progressive training

**Early Termination.** NSVF can represent transparent and solid objects equally well. However, for solid surfaces, the proposed volume rendering disperses the surface color along the ray, which means that it takes many unnecessary accumulation steps behind the surface to make the accumulated transparency $A(\boldsymbol{p}_0, \boldsymbol{v})$ reach 0. We therefore use a heuristic and stop evaluating points earlier when the accumulated transparency $A(\boldsymbol{p}_0, \boldsymbol{v})$ drops below a certain threshold $\epsilon$. In our experiments, we find that the setting $\epsilon = 0.01$ significantly accelerates the rendering process without causing any noticeable quality degradation.

### 3.3 Learning

Since our rendering process is fully differentiable, NSVF can be optimized end-to-end through back-propagation by comparing the rendered outputs with a set of target images, without any 3D supervision. To this end, the following loss is minimized:

$$\mathcal{L} = \sum_{(\boldsymbol{p}_0, \boldsymbol{v}) \in R} \|\boldsymbol{C}(\boldsymbol{p}_0, \boldsymbol{v}) - \boldsymbol{C}^*(\boldsymbol{p}_0, \boldsymbol{v})\|_2^2 + \lambda \cdot \Omega\left(A(\boldsymbol{p}_0, \boldsymbol{v})\right), \tag{5}$$

where $R$ is a batch of sampled rays, $\boldsymbol{C}^*$ is the ground-truth color of the camera ray, and $\Omega(.)$ is a beta-distribution regularizer proposed in Lombardi et al. (2019). Next, we propose a progressive training strategy to better facilitate learning and inference:

**Voxel Initialization** We start by learning implicit functions for an initial set of voxels subdividing an initial bounding box (with volume $V$) that roughly encloses the scene with sufficient margin. The initial voxel size is set to $l \approx \sqrt[3]{V/1000}$. If a coarse geometry (e.g. scanned point clouds or visual hull outputs) is available, the initial voxels can also be initialized by voxelizing the coarse geometry.

**Self-Pruning** Existing volume-based neural rendering works (Lombardi et al., 2019; Mildenhall et al., 2020) have shown that it is feasible to extract scene geometry on a coarse level after training. Based on this observation, we propose – *self-pruning* – a strategy to effectively remove non-essential voxels during training based on the coarse geometry information which can be further described using model's prediction on density. That is, we determine voxels to be pruned as follows:

$$V_i \text{ is pruned } \textbf{if } \min_{j=1...G} \exp(-\sigma(g_i\left(\boldsymbol{p}_j\right))) > \gamma, \ \ \boldsymbol{p}_j \in V_i, V_i \in \mathcal{V}, \tag{6}$$

where $\{\boldsymbol{p}_j\}_{j=1}^{G}$ are $G$ uniformly sampled points inside the voxel $V_i$ ($G = 16^3$ in our experiments), $\sigma(g_i\left(\boldsymbol{p}_j\right))$ is the predicted density at point $\boldsymbol{p}_j$, $\gamma$ is a threshold ($\gamma = 0.5$ in all our experiments). Since this pruning process does not rely on other processing modules or input cues, we call it *self-pruning*. We perform self-pruning on voxels periodically after the coarse scene geometry emerges.

**Progressive Training** The above pruning strategy enables us to progressively adjust voxelization to the underlying scene structure and adaptively allocate computational and memory resources to important regions. Suppose that the learning starts with an initial ray-marching step size $\tau$ and voxel size $l$. After certain steps of training, we halve both $\tau$ and $l$ for the next stage. Specifically, when halving the voxel size, we subdivide each voxel into $2^3$ sub-voxels and the feature representations of the new vertices (i.e. $\tilde{g}(.)$ in § 3.1) are initialized via trilinear interpolation of feature representations at the original eight voxel vertices. Note that, when using embeddings as voxel representations, we essentially increase the model capacity progressively to learn more details of the scene. In our experiments, we train synthetic scenes with 4 stages and real scenes with 3 stages. An illustration of self-pruning and progressive training is shown in Figure 2.

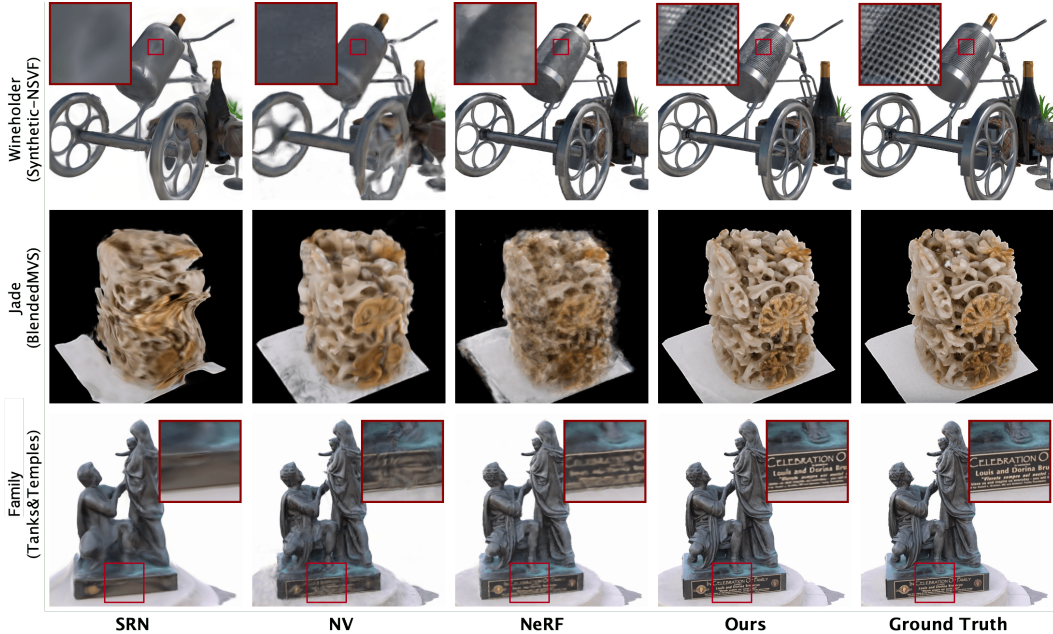

Figure 3: Comparisons on test views for scenes from the single-scene datasets. For *wineholder* and *family*, closeups are shown for clearer visual comparison.

## 4 Experiments

We evaluate the proposed NSVF on several tasks including multi-scene learning, rendering of dynamic and large-scale indoor scenes, and scene editing and composition. We also perform ablation studies to validate different kinds of feature representations and different options in progressive training. Please see the Appendix for more details on architecture, implementation, pre-processing of datasets and additional results. Please also refer to the supplemental video which shows the rendering quality.

### 4.1 Experimental Settings

**Datasets** (1) *Synthetic-NeRF:* The synthetic dataset used in Mildenhall et al. (2020) includes eight objects. (2) *Synthetic-NSVF:* We additionally render eight objects in the same resolution with more complex geometry and lighting effects. (3) *BlendedMVS:* We test on four objects from Yao et al. (2020). The rendered images are blended with the real images to have realistic ambient lighting.(4) *Tanks & Temples:* We evaluate on five objects from Knapitsch et al. (2017) where we use the images and label the object masks ourselves. (5) *ScanNet:* We use two real scenes from ScanNet (Dai et al., 2017). We extract both RGB and depth images from the original video.(6) *Maria Sequence:* This sequence is provided by Volucap with the meshes of 200 frames of a moving female. We render each mesh to create a dataset.

**Baselines** We adopt the following three recently proposed methods as baselines: Scene Representation Networks (SRN, Sitzmann et al., 2019b), Neural Volumes (NV, Lombardi et al., 2019), and Neural Radiance Fields (NeRF, Mildenhall et al., 2020), representing surface-based rendering, explicit and implicit volume rendering, respectively. See the Appendix for implementation details.

**Implementation Details** We model NSVF with a 32-dimentional learnable voxel embedding for each vertex, and apply positional encoding following (Mildenhall et al., 2020). The overall network architecture is shown in the Appendix Figure 9. For all scenes, we train NSVF using a batch size of 32 images on 8 Nvidia V100 GPUs, and for each image we sample 2048 rays. To improve training efficiency, we use a biased sampling strategy to only sample the rays which hits at least one voxel. For all the experiments, we prune the voxels periodically every 2500 steps and progressively halve the voxel and step sizes at 5k, 25k and 75k, separately. We have open-sourced our codebase at `https://github.com/facebookresearch/NSVF`

Table 1: The quantitative comparisons on test sets of four datasets. We use three metrics: PSNR ($\uparrow$), SSIM ($\uparrow$) and LPIPS ($\downarrow$) (Zhang et al., 2018) to evaluate the rendering quality. Scores are averaged over the testing images of all scenes, and we present the per-scene breakdown results in the Appendix. By default, NSVF is executed with early termination ($\epsilon = 0.01$). We also show results without using early termination ($\epsilon = 0$) denoted as NSVF$^0$.

| Models | Synthetic-NeRF | | | Synthetic-NSVF | | | BlendedMVS | | | Tanks and Temples | | |
|---|---|---|---|---|---|---|---|---|---|---|---|---|
| | PSNR | SSIM | LPIPS | PSNR | SSIM | LPIPS | PSNR | SSIM | LPIPS | PSNR | SSIM | LPIPS |
| SRN | 22.26 | 0.846 | 0.170 | 24.33 | 0.882 | 0.141 | 20.51 | 0.770 | 0.294 | 24.10 | 0.847 | 0.251 |
| NV | 26.05 | 0.893 | 0.160 | 25.83 | 0.892 | 0.124 | 23.03 | 0.793 | 0.243 | 23.70 | 0.834 | 0.260 |
| NeRF | 31.01 | 0.947 | 0.081 | 30.81 | 0.952 | 0.043 | 24.15 | 0.828 | 0.192 | 25.78 | 0.864 | 0.198 |
| NSVF$^0$ | **31.75** | **0.954** | 0.048 | **35.18** | **0.979** | **0.015** | 26.89 | **0.898** | 0.114 | **28.48** | **0.901** | 0.155 |
| NSVF | 31.74 | 0.953 | **0.047** | 35.13 | **0.979** | **0.015** | **26.90** | **0.898** | **0.113** | 28.40 | 0.900 | **0.153** |

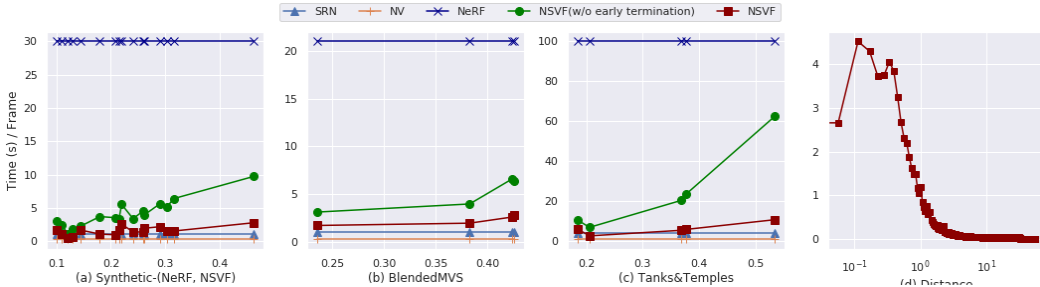

Figure 4: We report time taken to render one image for all the datasets in (a)-(c) where the x-axis stands for ascending foreground to background ratio and the y-axis for rendering time in second. We also show a plot curve for rendering time of NSVF on one synthetic scene in (d) when the camera is zooming out where the x-axis stands for the distance from the camera to the center of the object and the y-axis for rendering time in second.

## 4.2 Results

**Quality Comparison**  We show the qualitative comparisons in Figure 3. SRN tends to produce overly smooth rendering and incorrect geometry; NV and NeRF work better but are still not able to synthesize images as sharply as NSVF does. NSVF can achieve photo-realistic results on various kinds of scenes with complex geometry, thin structures and lighting effects.

Also, as shown in Table 1, NSVF significantly outperforms the three baselines on all the four datasets across all metrics. Note that NSVF with early termination ($\epsilon = 0.01$) produces almost the same quality as NSVF without early termination (denoted as NSVF$^0$ in Table 1). This indicates that early termination would not cause noticeable quality degradation while significantly accelerating computation, as will be seen next.

**Speed Comparison**  We provide speed comparisons on the models of four datasets in Figure 4 where we merge the results of *Synthetic-NeRF* and *Synthetic-NSVF* in the same figure considering their image sizes are the same. For our method, the average rendering time is correlated to the average ratio of foreground to background as shown in Figure 4 (a)-(c). That is because the higher the average ratio of foreground is, the more rays intersect with voxels. Thus, more evaluation time is needed. The average rendering time is also correlated to the number of intersected voxels. When a ray intersects a large number of voxels in the rendering of a solid object, early termination significantly reduces rendering time by avoiding many unnecessary accumulation steps behind the surface. These two factors can be seen in Figure 4 (d) where we show a zoom-out example.

For other methods, the rendering time is almost constant. This is because they have to evaluate all pixels with fixed steps, indicating a fixed number of points are sampled along each ray no matter whether the ray hits the scene or not, regardless of the scene complexity. In general, our method is around 10 $\sim$ 20 times faster than the state-of-the-art method NeRF, and gets close to SRN and NV.

**Storage Comparison**  The storage usage for the network weights of NSVF varies from 3.2 $\sim$ 16MB (including around 2MB for MLPs), depending on the number of used voxels (10 $\sim$ 100K). NeRF has two (coarse and fine) slightly deeper MLPs with a total storage usage of around 5MB.

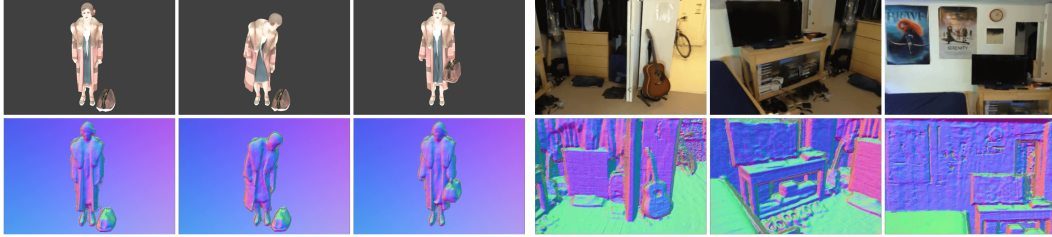

Figure 5: Our results on *Maria Sequence* (left) and *ScanNet* (right). We render testing trajectories and show three sampled frames in both RGB (top) and corresponding surface normals (below).

| Models | PSNR↑ | SSIM↑ | LPIPS↓ |
|---|---|---|---|
| NSVF | **32.04** | **0.965** | **0.020** |
| w/o POS | 30.89 | 0.954 | 0.043 |
| w/o EMB | 27.02 | 0.931 | 0.077 |
| w/o POS, EMB | 24.47 | 0.906 | 0.118 |

NSVF w/o EMB  NSVF w/o POS  NSVF  Ground Truth

Figure 7: The table on the left shows quantitative comparison of NSVF w/o positional encoding (w/o POS), w/o voxel embeddings (w/o EMB) or w/o both (w/o POS,EMB). The figure on the right shows visual comparison against the ground truth image.

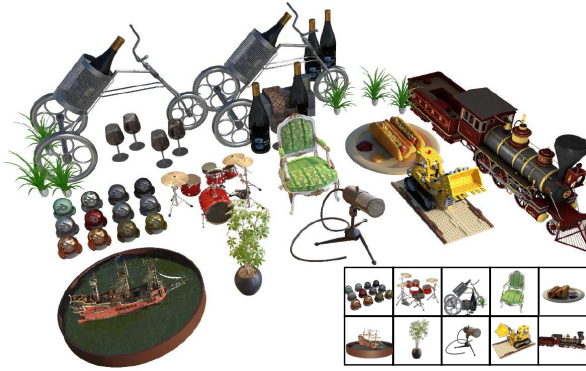

Figure 6: Scene composition and editing on the *Synthetic* datasets. The real images are presented bottom right.

**Rendering of Indoor Scenes & Dynamic Scenes** We demonstrate the effectiveness of our method on ScanNet dataset under challenging inside-out reconstruction scenarios. Our results are shown in Figure 5 where the initial voxels are built upon the point clouds from the depth images.

As shown in Figure 5, we also validate our approach on a corpus with dynamic scenes using the Maria Sequence. In order to accommodate temporal sequence with NSVF, we apply the hypernetwork proposed in Sitzmann et al. (2019b). We also include quantitative comparisons in the Appendix, which shows that NSVF outperforms all the baselines for both cases.

**Multi-scene Learning** We train a single model for all 8 objects from *Synthetic-NeRF* together with 2 additional objects (*wineholder*, *train*) from *Synthetic-NSVF*. We use different voxel embeddings for each scene while sharing the same MLPs to predict density and color. For comparison, we train NeRF model for the same datasets based on a hypernetwork (Ha et al., 2016). Without voxel embeddings, NeRF has to encode all the scene details with the network parameters, which leads to drastic quality degradation compared to single scene learning results. Table 2 shows that our method significantly outperforms NeRF on the multi-scene learning task.

Table 2: Results for multi-scene learning.

| Models | PSNR↑ | SSIM↑ | LPIPS↓ |
|---|---|---|---|
| NeRF | 25.71 | 0.891 | 0.175 |
| NSVF | **30.68** | **0.947** | **0.043** |

**Scene Editing and Scene Composition.** As shown in Figure 6, the learnt multi-object model can be readily used to compose more complex scenes by duplicating and moving voxels, and be rendered in the same way without overhead. Furthermore, our approach also supports scene editing by directly adjusting the presence of sparse voxels (See the re-composition of *wineholder* in Figure 6).

Table 3: Ablation for progressive training.

| R | PSNR↑ | SSIM↑ | LPIPS↓ | Speed (s) |
|---|---|---|---|---|
| 1 | 28.82 | 0.933 | 0.063 | 2.629 |
| 2 | 30.17 | 0.946 | 0.052 | 2.785 |
| 3 | 30.83 | 0.953 | 0.046 | 3.349 |
| 4 | **30.89** | **0.954** | **0.043** | 3.873 |

### 4.3 Ablation Studies

We use one object (*wineholder*) from the *Synthetic-NSVF* dataset which consists of parts with complex local patterns (grids) for ablation studies.

**Effect of Voxel Representations**   Figure 7 shows the comparison on different kinds of feature representations for encoding a spatial location. Voxel embeddings bring larger improvements to the quality than using positional encoding. Also, with both positional encoding and voxel embeddings, the model achieves the best quality, especially for recovering high frequency patterns.

**Effect of Progressive Training**   We also investigate different options for progressive training (see Table 3). Note that all the models are trained with voxel embeddings only. The performance is improved with more rounds of progressive training. But after a certain number of rounds, the quality improves only slowly while the rendering time increases. Based on this observation, our model performs 3-4 rounds of progressive training in the experiments.

## 5   Related Work

**Neural Rendering**   Recent works have shown impressive results by replacing or augmenting the traditional graphics rendering with neural networks, which is typically referred to as *neural rendering*. We refer the reader to recent surveys for neural rendering (Tewari et al., 2020; Kato et al., 2020).

- **Novel View Synthesis with 3D inputs:** DeepBlending (Hedman et al., 2018) predicts blending weights for the image-based rendering on a geometric proxy. Other methods (Thies et al., 2019; Kim et al., 2018; Liu et al., 2019a, 2020; Meshry et al., 2019; Martin Brualla et al., 2018; Aliev et al., 2019) first render a given geometry with explicit or neural textures into coarse RGB images or feature maps which are then translated into high-quality images. However, these works need 3D geometry as input and the performance would be affected by the quality of the geometry.

- **Novel View Synthesis without 3D inputs:** Other approaches learn scene representations for novel-view synthesis from 2D images. Generative Query Networks (GQN) (Eslami et al., 2018) learn a vectorized embedding of a 3D scene and render it from novel views. However, they do not learn geometric scene structure as explicitly as NSVF, and their renderings are rather coarse. Following-up works learned more 3D-structure aware representations and accompanying renderers (Flynn et al., 2016; Zhou et al., 2018; Mildenhall et al., 2019) with Multiplane Images (MPIs) as proxies, which only render a restricted range of novel views interpolating input views. Nguyen-Phuoc et al. (2018, 2019); Liu et al. (2019c) use a CNN-based decoder for differentiable rendering to render a scene represented as coarse-grained voxel grids. However, this CNN-based decorder cannot ensure view consistency due to 2D convolution kernels.

**Neural Implicit Representations.**   Implicit representations have been studied to model 3D geometry with neural networks. Compared to explicit representations (such as point cloud, mesh, voxels), implicit representations are continuous and have high spatial resolution. Most works require 3D supervision during training to infer the SDF value or the occupancy probability of any 3D point  (Michalkiewicz et al., 2019; Mescheder et al., 2019; Chen and Zhang, 2019; Park et al., 2019; Peng et al., 2020), while other works learn 3D representations only from images with differentiable renderers (Liu et al., 2019d; Saito et al., 2019, 2020; Niemeyer et al., 2019; Jiang et al., 2020).

## 6   Conclusion

We propose NSVF, a hybrid neural scene representations for fast and high-quality free-viewpoint rendering. Extensive experiments show that NSVF is typically over 10 times faster than the state-of-the-art (namely, NeRF) while achieving better quality. NSVF can be easily applied to scene editing and composition. We also demonstrate a variety of challenging tasks, including multi-scene learning, free-viewpoint rendering of a moving human, and large-scale scene rendering.

## 7 Broader Impact

NSVF provides a new way to learn a neural implicit scene representation from images that is able to better allocate network capacity to relevant parts of a scene. In this way, it enables learning representations of large-scale scenes at higher detail than previous approaches, which also leads to higher visual quality of the rendered images. In addition, the proposed representation enables much faster rendering than the state-of-the-art, and enables more convenient scene editing and compositing. This new approach to 3D scene modeling and rendering from images complements and partially improves over established computer graphics concepts, and opens up new possibilities in many applications, such as mixed reality, visual effects, and training data generation for computer vision tasks. At the same time it shows new ways to learn spatially-aware scene representations of potential relevance in other domains, such as object scene understanding, object recognition, robot navigation, or training data generation for image-based reconstruction.

The ability to capture and re-render, only from 2D images, models of real world scenes at very high visual fidelity, also enables the possibility to reconstruct and re-render humans in a scene. Therefore, any research on and practical application of this and all related reconstruction methods have to strictly respect personality rights and privacy regulations.

## Acknowledgments and Disclosure of Funding

We thank Volucap Babelsberg and the Fraunhofer Heinrich Hertz Institute for providing the *Maria dataset*. We also thank Shiwei Li, Nenglun Chen, Ben Mildenhall for the help with experiments; Gurprit Singh for discussion. Christian Theobalt was supported by ERC Consolidator Grant 770784. Lingjie Liu was supported by Lise Meitner Postdoctoral Fellowship. The computational work for this article was partially performed on resources of the National Supercomputing Centre, Singapore (https://www.nscc.sg).

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
