[Supplementary Material]

# A    Additional Details of the Method

## A.1    Algorithm

We present the algorithm of rendering with NSVF as follows in Algorithm 1. We additionally return the transparency $A$, and the expected depth $Z$ which can be further used for visualizing the normal with finite difference.

---

**Algorithm 1: Neural Rendering with NSVF**

**Input:**  camera $p_0$, ray direction $v$, step size $\tau$, threshold $\epsilon$, voxels $\mathcal{V} = \{V_1, \ldots, V_K\}$, background $c_{\mathrm{bg}}$,
background maximum depth $z_{\max}$, parameters of the MLPs $\theta$
**Initialize:** transparency $A = 1$, color $C = 0$, expected depth $Z = 0$
**Ray-voxel Intersection:** Return all the intersections of the ray with $k$ intersected voxels, sorted from near to far: $z_{t_1}^{\mathrm{in}}, z_{t_1}^{\mathrm{out}}, \ldots, z_{t_k}^{\mathrm{in}}, z_{t_k}^{\mathrm{out}}$, where $\{t_1, \ldots, t_k\} \subset \{1 \ldots K\}, k < K$;
**if** $k > 0$ **then**
    **Stratified sampling:** $z_1, \ldots, z_m$ with step size $\tau$, where $z_1 \geq z_{t_1}^{\mathrm{in}}$ and $z_m \leq z_{t_k}^{\mathrm{out}}$;
    **Include voxel boundaries:** $\tilde{z}_1, \ldots \tilde{z}_{2k+m} \leftarrow \mathrm{sort}\left(z_1, \ldots, z_m; z_{t_1}^{\mathrm{in}}, z_{t_1}^{\mathrm{out}}, \ldots, z_{t_k}^{\mathrm{in}}, z_{t_k}^{\mathrm{out}}\right)$;
    **for** $j \leftarrow 1$ **to** $2k + m - 1$ **do**
        **Obtain midpoints and intervals:** $\hat{z}_j \leftarrow \frac{\tilde{z}_j + \tilde{z}_{j+1}}{2}, \Delta_j \leftarrow \tilde{z}_{j+1} - \tilde{z}_j$;
        **if** $A > \epsilon$ **and** $\Delta_j > 0$ **and** $p(\hat{z}_j) \in V_i (\exists i \in \{t_1, \ldots, t_k\})$ **then**
            $\alpha \leftarrow \exp\left(-\sigma_\theta\left(g_i(p(\hat{z}_j))\right) \cdot \Delta_j\right), \quad c \leftarrow c_\theta\left(g_i(p(\hat{z}_j)), v\right)$;
            $C \leftarrow C + A \cdot (1 - \alpha) \cdot c, \quad Z \leftarrow Z + A \cdot (1 - \alpha) \cdot \hat{z}_j, \quad A \leftarrow A \cdot \alpha$;

$C \leftarrow C + A \cdot c_{\mathrm{bg}}, \quad Z \leftarrow Z + A \cdot z_{\max}$;
**Return:** $C, Z, A$

---

## A.2    Overall Pipeline

We present the illustrations of the overall pipeline to better demonstrate the proposed approach in Figure 8. For any given camera position $p_0$ and the ray direction $v$, we render its color $c$ with NSVF by first intersecting the ray with a set of sparse voxels, and sampling and accumulating the color and density inside every intersected voxel, which are predicted by neural networks.

Figure 8: Illustration of the differenitable volume rendering procedure with NSVF. For any given camera position $p_0$ and the ray direction $v$, we first intersect the ray with a set of sparse voxels, then predict the colors and densities with neural networks for points sampled along the ray inside voxels, and accumulate the colors and densities of the sampled points to get the rendered color $C(p_0, v)$.

# B    Additional Experimental Settings

## B.1    Datasets

We present more details about the datasets we used. We conduct the experiments of single-scene learning on five datasets, including three synthetic datasets and two real datasets:

- *Synthetic-NeRF.* We use the NeRF (Mildenhall et al., 2020) synthetic dataset which includes eight objects rendered with path tracing. Each object is rendered to produce 100 views for training and 200 for testing at $800 \times 800$ pixels.

- *Synthetic-NSVF.* To demonstrate the ability of NSVF to handle various conditions, we additionally render eight objects in $800 \times 800$ with more complex geometry and lighting effects. Details on the original source files and license information are given below:

  - Wineholder(CC-0) `https://www.blendswap.com/blend/15899`
  - Steamtrain(CC-BY-NC) `https://www.blendswap.com/blend/16763`
  - Toad(CC-0) `https://www.blendswap.com/blend/13078`
  - Robot(CC-BY-SA) `https://www.blendswap.com/blend/10597`
  - Bike(CC-BY) `https://www.blendswap.com/blend/8850`
  - Palace(CC-BY-NC-SA) `https://www.blendswap.com/blend/14878`
  - Spaceship(CC-BY) `https://www.blendswap.com/blend/5349`
  - Lifestyle(CC-BY) `https://www.blendswap.com/blend/8909`

- *BlendedMVS.* We test on four objects of a recent synthetic MVS dataset, BlendedMVS (Yao et al., 2020) [2]. The rendered images are blended with the real images to have realistic ambient lighting. The image resolution is $768 \times 576$. One eighth of the images are held out as test sets.

- *Tanks & Temples.* We evaluate on five objects of Tanks and Temples (Knapitsch et al., 2017) [3] real scene dataset. We label the object masks ourselves with the software of Altizure [4], and sample One eighth of the images for testing. The image resolution is $1920 \times 1080$.

- *ScanNet.* We use two real scenes of an RGB-D video dataset for large-scale indoor scenes, ScanNet (Dai et al., 2017)[5]. We extract both the RGB and depth images of which we randomly sample $20\%$ as training set and use the rest for testing. The image is scaled to $640 \times 480$.

For the multi-scene learning, we show our result of training with all the scenes of *Synthetic-NeRF* and two out of *Synthetic-NSVF*, and the result of training with all the frames of a moving human:

- *Maria Sequence.* This sequence is provided by *Volucap* with the meshes of 200 frames of a moving female. We render each mesh from 50 viewpoints sampled on the upper hemisphere at $1024 \times 1024$ pixels. We also render 50 additional views in a circular trajectory as the test set.

## B.2 Implementation Details

**Architecture** The proposed model assigns a 32-dimentional learnable voxel embedding to each vertex, and applies positional encoding with maximum frequency as $L = 6$ (Mildenhall et al., 2020) to the feature embedding aggregated by eight voxel embeddings of the corresponding voxel via trilinear interpolation. As a comparison, we also train our model without positional encoding where we set the voxel embedding dimension $d = 416$ in order to have comparable feature vectors as the complete model. We use around $1000$ initial voxels for each scene. The final number of voxels after pruning and progressive training varies from 10k to 100k (the exact number of voxels differs scene by scene due to varying sizes and shapes), with an effective number of $0.32 \sim 3.2$M learnable parameters in our default voxel embedding settings.

The overall network architecture of our default model is illustrated in Figure 9 with $\sim 0.5$M parameters, not including voxel embeddings. Note that, our implementation of the MLP is slightly shallower than many of the existing works (Sitzmann et al., 2019b; Niemeyer et al., 2019; Mildenhall et al., 2020). By utilizing the voxel embeddings to store local information in a distributed way, we argue that it is sufficient to learn a small MLP to gather voxel information and make accurate predictions.

Figure 9: A visualization of the proposed NSVF architecture. For any input $(\boldsymbol{p}, \boldsymbol{v})$, the model first obtains the feature representation by querying and interpolating the voxel embeddings with the 8 corresponding voxel vertices, and then uses the computed feature to further predicts $(\sigma, \boldsymbol{c})$ using a MLP shared by all voxels.

**Training & Inference**   We train NSVF using a batch size of 32 images on 8 Nvidia V100 GPUs, and for each image we sample 2048 rays. To improve training efficiency, we use a biased sampling strategy to sample rays where it hits at least one voxel. We use Adam optimizer with an initial learning rate of $0.001$ and linear decay scheduling. By default, we set the step size $\tau = l/8$, while the initial voxel size ($l$) is determined as discussed in § 3.3.

For all experiments, we prune the voxels with Eq (6) periodically for every 2500 steps. All our models are trained with $100 \sim 150$k iterations by progressively halving the voxel and step sizes at 5k, 25k and 75k, separately. At inference time, we use the threshold of $\epsilon = 0.01$ for early termination for all models. As a comparison, we also conduct experiments without setting up early termination. Our model is implemented in PyTorch using Fairseq framework[6].

**Evaluation**   We measure the quality on test sets with three metrics: PSNR, SSIM and LPIPS (Zhang et al., 2018). For the comparisons in speed, we render NSVF and the baselines with one image per batch and calculate the average rendering time using a single Nvidia V100 GPU.

**Multi-scene Learning**   Our experiments also require learning NSVF on multiple objects where a voxel location may be shared by different objects. In this work, we present two ways to tackle this issue. First, we use the navie approach that learns saperate embedding matrices for each object and only the MLP are shared. This is well suitable when the categories of target objects are quite distinct, and this can essentially increase the model capacity by extending the number of embeddings infinitely. We validate this method for the multi-scene learning task on all 8 scenes from *Synthetic-NeRF* together with 2 additional scenes (*wineholder, train*) from *Synthetic-NSVF*.

However, when modeling multiple objects that have similarities (e.g., a class of objects, or a moving sequence of the target object), it is more suitable to have shared voxel representations. Here we learn a set of voxel embeddings for each voxel position, while maintaining a unique embedding vector for each object. We compute the final voxel representation based on hypernetworks (Sitzmann et al., 2019b) with the object embedding as the input. We show our results on *Maria Sequence*.

### B.3   Additional Baseline Details

**Scene Representation Networks (SRN, Sitzmann et al., 2019b)**   We use the original code open-sourced by the authors [7]. To enable training on higher resolution images, we employ the ray-based sampling strategy that is similarly used in neural volumes and NeRF. We use the batch size of 8 and 5120 rays per image. We found that clipping gradient norm to 1 greatly improves stability during training. All models are trained for 300k iterations.

Figure 10: Additional examples and comparisons sampled from *Synthetic-NSVF*, *BlendedMVS* and *Tanks&Temples* datasets. Please see more results in the supplemental video.

Figure 11: An example of zooming in and out without any visible artifacts
.

**Neural Volumes (NV, Lombardi et al., 2019)**    We use the original code opensourced by the authors [8]. We use batch size of 8 and $128 \times 128$ rays per image. The center and scale of each scene are determined using the visual hull to place the scene within a cube that spans from -1 to 1 on each axis, as required by implementation. All models are trained for 40k iterations.

**Neural Radiance Fields (NeRF, Mildenhall et al., 2020)**    We use the NeRF code opensourced by the authors [9] and train on a single scene with the default settings used in NeRF with 100k-150k iterations. We scale the bounding box of each scene used in NSVF so that the bounding box lies within a cube of side length 2 centered at origin. To train on multiple scenes, we employ the auto-decoding scheme using a hypernetwork as described in SRN (Sitzmann et al., 2019b). We use a 1-layer hypernetwork to predict weights for all the scenes. The latent code dimension is 256.

# C    Additional Results

## C.1    Per-scene breakdown

We show the per-scene breakdown analysis of the quantitative results presented in the main paper (Table 1) for the four datasets (*Synthetic-NeRF*, *Synthetic-NSVF*, *BlendedMVS* and *Tanks&Temples*). Table 4 reports the comparisons with the three baselines in three metrics. Our approach achieves the best performance on both PSNR and LPIPS metrics across almost all the scenes, especially for datasets with real objects.

## C.2    Additional Examples

In Figure 10, we present additional examples for individual scenes not shown in the main paper. We would like to highlight how well our method performs across a wide variety of scenes, showing much better visual fidelity than all the baselines.

## C.3    Additional Analysis

**Effects of Voxel Sizes.**    In Table 5, we show additional comparison on *wineholder* where we fix the ray marching step size as the initial values, while training the model with different voxel sizes. The first column shows the ratio compared to the initial voxel size. It is clear that reducing the voxel size helps improve the rendering quality, indicating that progressively increasing the model's capacity alone helps model details better for free-viewpoint rendering.

**Geometry Reconstruction Accuracy**    We would like to expand on the observation that we have briefly touched on in the main paper regarding the nature of surface-based and volume-based renderers.

Table 4: Detailed breakdown of quantitative metrics of individual scenes for all 4 datasets for our method and 3 baselines. All scores are averaged over the testing images.

| | Chair | Drums | Lego | Mic | Materials | Ship | Hotdog | Ficus |
|---|---|---|---|---|---|---|---|---|
| **Synthetic-NeRF** | | | | | | | | |
| | | | | PSNR↑ | | | | |
| SRN | 26.96 | 17.18 | 20.85 | 26.85 | 18.09 | 20.60 | 26.81 | 20.73 |
| NV | 28.33 | 22.58 | 26.08 | 27.78 | 24.22 | 23.93 | 30.71 | 24.79 |
| NeRF | 33.00 | 25.01 | **32.54** | 32.91 | 29.62 | 28.65 | 36.18 | 30.13 |
| Ours | **33.19** | **25.18** | 32.29 | **34.27** | **32.68** | **27.93** | **37.14** | **31.23** |
| | | | | SSIM↑ | | | | |
| SRN | 0.910 | 0.766 | 0.809 | 0.947 | 0.808 | 0.757 | 0.923 | 0.849 |
| NV | 0.916 | 0.873 | 0.880 | 0.946 | 0.888 | 0.784 | 0.944 | 0.910 |
| NeRF | 0.967 | 0.925 | **0.961** | 0.980 | 0.949 | **0.856** | 0.974 | 0.964 |
| Ours | **0.968** | **0.931** | 0.960 | **0.987** | **0.973** | 0.854 | **0.980** | **0.973** |
| | | | | LPIPS↓ | | | | |
| SRN | 0.106 | 0.267 | 0.200 | 0.063 | 0.174 | 0.299 | 0.100 | 0.149 |
| NV | 0.109 | 0.214 | 0.175 | 0.107 | 0.130 | 0.276 | 0.109 | 0.162 |
| NeRF | 0.046 | 0.091 | 0.050 | 0.028 | 0.063 | 0.206 | 0.121 | 0.044 |
| Ours | **0.043** | **0.069** | **0.029** | **0.010** | **0.021** | **0.162** | **0.025** | **0.017** |

| | Wineholder | Steamtrain | Toad | Robot | Bike | Palace | Spaceship | Lifestyle |
|---|---|---|---|---|---|---|---|---|
| **Synthetic-NSVF** | | | | | | | | |
| | | | | PSNR↑ | | | | |
| SRN | 20.74 | 25.49 | 25.36 | 22.27 | 23.76 | 24.45 | 27.99 | 24.58 |
| NV | 21.32 | 25.31 | 24.63 | 24.74 | 26.65 | 26.38 | 29.90 | 27.68 |
| NeRF | 28.23 | 30.84 | 29.42 | 28.69 | 31.77 | 31.76 | 34.66 | 31.08 |
| Ours | **32.04** | **35.13** | **33.25** | **35.24** | **37.75** | **34.05** | **39.00** | **34.60** |
| | | | | SSIM↑ | | | | |
| SRN | 0.850 | 0.923 | 0.822 | 0.904 | 0.926 | 0.792 | 0.945 | 0.892 |
| NV | 0.828 | 0.900 | 0.813 | 0.927 | 0.943 | 0.826 | 0.956 | 0.941 |
| NeRF | 0.920 | 0.966 | 0.920 | 0.960 | 0.970 | 0.950 | 0.980 | 0.946 |
| Ours | **0.965** | **0.986** | **0.968** | **0.988** | **0.991** | **0.969** | **0.991** | **0.971** |
| | | | | LPIPS↓ | | | | |
| SRN | 0.224 | 0.082 | 0.204 | 0.120 | 0.075 | 0.240 | 0.061 | 0.120 |
| NV | 0.204 | 0.121 | 0.192 | 0.096 | 0.067 | 0.173 | 0.056 | 0.088 |
| NeRF | 0.096 | 0.031 | 0.069 | 0.038 | 0.019 | 0.031 | 0.016 | 0.047 |
| Ours | **0.020** | **0.010** | **0.032** | **0.007** | **0.004** | **0.018** | **0.006** | **0.020** |

| | Jade | Fountain | Char | Statues | Ignatius | Truck | Barn | Cate | Family |
|---|---|---|---|---|---|---|---|---|---|
| **BlendedMVS** | | | | | **Tanks& Temple** | | | | |
| | | | | PSNR↑ | | | | | |
| SRN | 18.57 | 21.04 | 21.98 | 20.46 | 26.70 | 22.62 | 22.44 | 21.14 | 27.57 |
| NV | 22.08 | 22.71 | 24.10 | 23.22 | 26.54 | 21.71 | 20.82 | 20.71 | 28.72 |
| NeRF | 21.65 | 25.59 | 25.87 | 23.48 | 25.43 | 25.36 | 24.05 | 23.75 | 30.29 |
| Ours | **26.96** | **27.73** | **27.95** | **24.97** | **27.91** | **26.92** | **27.16** | **26.44** | **33.58** |
| | | | | SSIM↑ | | | | | |
| SRN | 0.715 | 0.717 | 0.853 | 0.794 | 0.920 | 0.832 | 0.741 | 0.834 | 0.908 |
| NV | 0.750 | 0.762 | 0.876 | 0.785 | 0.922 | 0.793 | 0.721 | 0.819 | 0.916 |
| NeRF | 0.750 | 0.860 | 0.900 | 0.800 | 0.920 | 0.860 | 0.750 | 0.860 | 0.932 |
| Ours | **0.901** | **0.913** | **0.921** | **0.858** | **0.930** | **0.895** | **0.823** | **0.900** | **0.954** |
| | | | | LPIPS↓ | | | | | |
| SRN | 0.323 | 0.291 | 0.208 | 0.354 | 0.128 | 0.266 | 0.448 | 0.278 | 0.134 |
| NV | 0.292 | 0.263 | 0.140 | 0.277 | 0.117 | 0.312 | 0.479 | 0.280 | 0.111 |
| NeRF | 0.264 | 0.149 | 0.149 | 0.206 | 0.111 | 0.192 | 0.395 | 0.196 | 0.098 |
| Ours | **0.094** | **0.113** | **0.074** | **0.171** | **0.106** | **0.148** | **0.307** | **0.141** | **0.063** |

Table 5: Effect of voxel size on the *wineholder* test set.

| Voxel | PSNR↑ | SSIM↑ | LPIPS↓ | Speed (s/frame) |
|-------|-------|-------|--------|-----------------|
| 1 | 28.82 | 0.933 | 0.063 | 2.629 |
| 1/2 | 29.22 | 0.938 | 0.057 | 1.578 |
| 1/4 | 29.70 | 0.944 | 0.052 | **1.369** |
| 1/8 | **30.17** | **0.948** | **0.047** | 1.515 |

As we have mentioned, surface-based rendering methods (e.g. SRN) require an accurate surface to be able to learn the color well. A failure case where geometry fails to be learnt is seen in the "Character" scene in Figure 10. In addition, we observe that SRN frequently gets stuck in a local minima so that the geometry is incorrect but is nevertheless approximately multi-view consistent. We find that this phenomenon occurs much less frequently in volume rendering methods including ours. NV, due to limited spatial resolution, is unable to capture high frequency details. NeRF generally works well while is still not able to synthesize images as sharply as NSVF does. Furthermore, NeRF suffers from a slow rendering process due to its inefficient sampling strategy. For instance. it takes 30s to render an $800 \times 800$ image with NeRF.

**Zoom-In & -Out**  Our model naturally supports zooming in and out for a trained object. We show the results in Figure 11.

Table 6: Quantitative results on *Wineholder* of NSVF with different threshold $\epsilon$ for early termination.

| $\epsilon$ | PSNR↑ | SSIM↑ | LPIPS↓ | Speed (s/frame) |
|-----------|-------|-------|--------|-----------------|
| 0.000 | 31.93 | 0.965 | 0.021 | 4.0 |
| 0.001 | 32.03 | 0.965 | 0.020 | 2.1 |
| 0.010 | **32.04** | **0.965** | **0.020** | 2.0 |
| 0.100 | 29.99 | 0.947 | 0.029 | 1.7 |

Table 7: Comparison of one-round training and progressive training on *Wineholder*.

| Method | PSNR↑ | SSIM↑ | LPIPS↓ |
|--------|-------|-------|--------|
| One-round | 29.77 | 0.946 | 0.033 |
| Progressive | **32.04** | **0.965** | **0.020** |

**Effect of Early Termination**  The quantitative results on *Wineholder* of NSVF with early termination with different thresholds are shown in Table 6. The selection of $\epsilon = 0.01$ gives the best trade-off between quality and rendering speed.

**Comparion with one round of training at the final resolution**  As shown in Table 7, our test on *Wineholder* shows that compared with one-round training, our progressive training is faster and easier to train, uses less space and achieves better quality.

### C.4   Details for Experiments on ScanNet

We list the details of learning on the ScanNet dataset. We first extract point clouds from all the RGBD images using known camera poses, and register them in the same 3D space. We then initialize a voxel based on the extracted points instead of using a bounding box. No pruning or progressive training are applied in this case. Furthermore, we integrate an additional depth loss based on the provided depth image, that is,

$$\mathcal{L}_{\text{depth}} = \sum_{\boldsymbol{p}_0, \boldsymbol{v}} |Z(\boldsymbol{p}_0, \boldsymbol{v}) - Z^*(\boldsymbol{p}_0, \boldsymbol{v})|_1 \tag{7}$$

where $Z^*$ is the ground truth depth and $Z$ is the expected distance where each ray terminates at this distance in Algorithm 1. We show more qualitative results in Figure 12  and the quantitative comparisons in Table 8 where NSVF achieves the best performance.

Table 8: Quantitative results for *ScanNet (one scene)* (Left) and *Maria Sequence* (Right). Here geometry accuracy is measured by RMSE of ground-truth depths and depths of rendered geometry. Note that no result for NV is reported for *ScanNet* because training failed to converge.

| | RMSE↓ | PSNR↑ | SSIM↑ | LPIPS↓ | | PSNR↑ | SSIM↑ | LPIPS↓ |
|---|---|---|---|---|---|---|---|---|
| SRN | 14.764 | 18.25 | 0.592 | 0.586 | SRN | 29.12 | 0.969 | 0.036 |
| NeRF | 0.681 | 22.99 | 0.620 | 0.369 | NV | 33.86 | 0.979 | 0.027 |
| Ours (w/o depth) | 0.210 | 25.07 | 0.668 | 0.315 | NeRF | 34.19 | 0.980 | 0.026 |
| Ours (w/ depth) | **0.079** | **25.48** | **0.688** | **0.301** | Ours | **38.92** | **0.991** | **0.010** |

Figure 12: Our sampled results on ScanNet of two different rooms. From left to right: the predicted image, the initial voxels from the point clouds, and the predicted geometry normals.
.

## C.5 Details for Experiments on Maria Sequence

We present additional details for learning on the Maria sequence. The Maria sequence consists of 200 frames of different poses if the same character. Since there exists strong correlation from frame to frame, we model all frames with the same set of initial voxels (a bounding box covers all 200 frames) and utilize a hypernetwork described in (Sitzmann et al., 2019b) to output the weights of the MLPs with the frame index as inputs. We also conduct quantitative comparisons on testing views, and report scores in Table 8. NSVF significantly outperforms all the baseline approaches.

## C.6 Procedure for Scene Editing and Composition

The learnt NSVF representations can be readily used for editing and composition. We show the basic procedure to edit a real scene in the following three steps: (1) learn and extract sparse voxels with multi-view 2D input images; (2) apply editing (e.g. translation, cloning, removal, etc.) on the voxels; (3) read the modified voxels and render new images. We illustrate the procedure in Figure 13. Furthermore, by learning the model with multiple objects, we can easily render composed scenes by rearranging learned voxels and rendering at the same time.

Figure 13: Illustration of scene editing and rendering with NSVF.

# D Limitations and Future Work

Although NSVF can efficiently generate high-quality novel views and significantly outperform existing methods, there are three major limitations:

(i) Our method cannot handle scenes with complex background. We assume a simple constant background term ($c_{bg}$). However, real scenes usually have different backgrounds when viewed from

different points. This makes it challenging to capture their effects correctly without the interference on the learning of the target scenes.

(ii) We set the threshold for self-pruning to be 0.5 in all the experiments. Although this works well for general scenes, incorrect pruning may occur for very thin structures if the threshold is not set properly.

Figure 14: A comparison between NSVF output (left) and the groud-truth (right) of a cropped view sampled from scene *lifestyle* (*Synthetic-NSVF* dataset).

(iii) Similar to Sitzmann et al. (2019b); Mildenhall et al. (2020), NSVF learns the color and density as a "black-box" function of the query point location and the camera-ray direction. Therefore, the rendering performance highly depends on the distribution of training images, and may produce severe artifacts when the training data is insufficient or biased for predicting complex geometry, materials and lighting effects (see Figure 14 where the refraction on the glass bottle is not learnt correctly). A possible future direction is to incorporate the traditional radiance and rendering equation as a physical inductive bias into the neural rendering framework. This can potentially improve the robustness and generalization of the neural network models.

(iv) The current learning paradigm requires known camera poses as inputs to initialize rays and their direction. For real world images, there is currently no mechanism to handle unavoidable errors in camera calibration. When our target data consists of single-view images of multiple objects, it is even more difficult to obtain accurately registered poses in real applications. A promising avenue for future research would be to use unsupervised techniques such as GANs (Nguyen-Phuoc et al., 2019) to simultaneously predict camera poses for high-quality free-viewpoint rendering results.

## Footnotes

[2]`https://github.com/YoYo000/BlendedMVS`

[3]`https://tanksandtemples.org/download/`

[4]`https://github.com/altizure/altizure-sdk-offline`

[5]`http://www.scan-net.org/`

[6]https://github.com/pytorch/fairseq

[7]https://github.com/vsitzmann/scene-representation-networks

[8]`https://github.com/facebookresearch/neuralvolumes`

[9]`https://github.com/bmild/nerf`