[Reviews · NeurIPS 2020]

Review 1

Summary and Contributions: This paper proposes a new method called Neural Sparse Voxel Fields (NSVF for short), which combines sparse voxel representation and implicit function representation to encode the scene and render new views. The authors show that by sparse voxel structure, the ray sampling points can be largely reduced and the new view can be rendered in a much faster speed. Also, by implicit function, the method achieves better rendering effect than recent papers.

Strengths: + oct tree structure results in faster rendering speed While NERF has shown a similar idea of sampling more points at dense region, this paper adopts explicit sparse octree structure voxel to avoid points samping in void region and only sample rays in necessary regions. This results in much faster rendering time. + explicit geometry + scene editing A sparse voxel representation could already give us a rough scene geometry, which nerf and neural volume need further process to get. This also enables scene editing by cutting and pasting the optimizated voxel grids. + better results Since NSVF only focuses on necessary regions, it is more compact representation and the authors shows it get better results than NERF and traditional Neural Volume.

Weaknesses: I didn't find too big limitations. The method is a little complicated and it would be very helpful if the authors release the code. The scene editing is not very clear. You are optimizing the voxel representation and MLP together. When you do the scene editing, you can merge two voxels, but when you do the rendering, how do you merge MLP layers?(I assume that each scene has its own MLP)? Better network size comparison. The author mentioned that in NSVF a much smaller MLP is needed, but compared to NERF, the extra voxel features is needed. It is better to compare the whole size rather than just MLP size.

Correctness: Yes.

Clarity: Yes.

Relation to Prior Work: Yes.

Reproducibility: Yes

Additional Feedback: I enjoy reading this paper, which is written in a clear way. This paper merges the advantages of the voxel representation and implicit function representation, and employ octree to largely speed up the rendering time. The progressive training policy produces very promising results. To further strength then paper, I would suggest the author do more analysis on the model size comparison and clarify the scene editing part. After rebuttal: Thanks for the rebuttal. My concerns are addressed.


Review 2

Summary and Contributions: This work proposes a neural scene representation based on the sparse voxel octree structure.

Strengths: The octree representation has several advantages: Each voxel has a implicit field and thus it has the better capacity to represent the scene and better capability to represent local details. Since it is based on the octree representation, it is natural to prune the empty voxels and thus leads to faster inferece during rendering. The experiment is thorough and clearly shows it is better than SOTA in three commonly used image quality metrics. The visual results are very good, I especially like the interactive demo in the video.

Weaknesses: It is straightforward to evolve from regular voxel representation to octree representation given similar evolution has happened first in computer graphics and recently in deep learning based 3D reconstruction; this prevents this work from being the top works in this field (Though I think it is still a good work.) The storage usage is not discussed in this work. I expect more storage might be used to represent a scene comparing to the other works or traditional formats (like 3D point cloud or mesh). I think it worth a discussion.

Correctness: Mostly correct.

Clarity: It is not clear to me which scenes are recovered from rgbd images since all the datasets seems to have 3D shapes. I think it is important to compare the results with and w/o the voxel initialization from existing 3D shape information, because the work mentioned at several places that it applies to RGB images. In Figure 3, I don't understand how does the ratio of foreground to background is computed. Is it based on the image pixels? Could it go to 1. How does the size of voxels affects rendering speed? I am also curious why this work can recover very small details for the synthetic scenes (like thin structures in Figure 4 Train) but not for the real world scene (like the tire textures in Figure 4 Caterpillar). Is it because there are almost infinite data for the synthetic scenes?

Relation to Prior Work: Clear. This most recent work `neural point cloud rendering via multi-plane projection` can be cited and dicussed.

Reproducibility: Yes

Additional Feedback: After rebuttal: After reading the rebuttal and other reviews, I am still positive about this work.


Review 3

Summary and Contributions: The authors propose a method for learning a neural scene representation from a set of posed RGB images that can be used to render images from novel viewpoints. They propose a local feature representation based on a sparse octree-based voxel grid. An MLP is then used to render the density and the color from this feature representation. For modeling the rendering process, they leverage the model proposed in Mildenhall et al. The neural scene representation is optimized using a progressive learning scheme that iteratively refines the voxel resolution if it is needed. The main contribution is to use a sparse octree-based voxel grid as the data structure and locally store the feature representation to represent the scene. Furthermore, the progressive training scheme to learn the representation from coarse to fine leveraging the octree-structure of the scene representation. The method is evaluated on synthetic as well as real-world datasets. Furthermore, they show potential applications such as representation of dynamic scenes and scene editing and scene composition.

Strengths: - The combination of an octree-based data structure with MLPs significantly improves the runtime of the neural ray-casting and shows impressive results (quantitative and qualitative). - The authors clearly relate their method to existing methods by showing that the existing methods are a special case of their more general method. - They thoroughly evaluate their method on three different datasets. - Furthermore, they performed various experiments with different settings (dynamic scenes, large-scale scenes, scene composition and multi-object training) show-casing the applicability of the proposed method.

Weaknesses: - The contribution of local voxel-bounded implicit fields is not motivated and evaluated enough. The authors showed that voxel embeddings improve the performance. However, an experiment that shows the performance with different resolutions (partly done with the progressive training evaluation) would improve the paper. - Similarly, the contribution of progressive training is not evaluated enough. In order to be a valid contribution, the authors need to show that the progressive training is not only better than one round of training at the initial resolution but also better than one round of training at the increased resolution. As it is done now (in the ablation study), there is the possibility that the improved performance is due to the higher voxel resolution. - For the experiments on large-scale scenes and dynamic scenes, qualitative results for existing methods are missing. Furthermore, quantitative results are missing for these experiments (also for existing methods). - The geometric result in the ScanNet experiment does not seem to be very good. Therefore, it would be interesting to see how accurate the represented geometry is and how it affects the overall performance of the method. - The effect of the early termination is not properly evaluated in the paper. - The initial grid resolution seems to be already very high. It would be interesting to see how this affects the method. - There should also be comparisons to DeepVoxels since this work also leverages local voxels as a feature representation.

Correctness: - Equation 2 seems to be incorrect as it is written and cited in the paper. In Mildenhall et al. there is a summation and not a product over the transmittance probability along the ray. - The claim that the proposed method is 10 times faster than state of the art is very strong given the results in the Figure 3. According to this figure, this is heavily dependent on the scene and the view to render. The authors should consider reformulating.

Clarity: - Some parts are not very clear in the paper. The different experimental settings should be better explained in the main paper. Especially, the paragraphs on rendering of dynamic scenes and multi-object learning would benefit from rewriting. - Furthermore, some paragraphs are not carefully written (e.g. effect of progressive training). Some improvements on the writing (spelling, grammar, and word usage) need to be done. - The authors stress out the benefit of sparse voxel grids (contribution #2). However, I cannot find a clear description of this process in the main paper. If it's listed as a contribution, it should be clearly described.

Relation to Prior Work: - The relation to Mildenhall et al. and Lombardi et al. is well communicated in the paragraph 'special cases'. - The related work section is very brief given the size of this field. Some related works that are missing: - Novel View Synthesis: Texture Fields - Furthermore, there is a whole line of work missing regarding implicit neural representations: OccupancyNetworks, DeepSDF, IM-Net, DSIN

Reproducibility: Yes

Additional Feedback: - The explanation of the evaluation metrics is missing. There should be some information about the different aspects they quantify (e.g. the authors should discuss why NeRF is better in SSIM). - How can overfitting be visible in the results? All methods are overfitted to the training scenes? Or do you mean overfitting to the training viewpoints? - The paper needs some rewriting and some checks on the grammar: - L19: We - L137: space - Caption Figure 5: trajectories - L253: performation --> performance - Are all methods optimized using the same views? Or are pretrained models used for SRN, NV and NeRF? - What's the reason that (almost) everything is represented perfectly in the locomotive scene, but there are clear differences between the proposed method and the ground-truth for the other datasets?


Review 4

Summary and Contributions: This paper presents a method to learn a volumetric representation of a scene for free-viewpoint rendering. Different from previous methods that use dense voxel grids, this paper applies sparse voxel grids based on an octree architecture. Such an architecture enables the method to learn volumes of higher resolution and achieve faster rendering.

Strengths: 1. The application of a sparse octree representation is inspiring and tackles an important limitation of previous volume-based methods. Such a representation is efficient and enables faster rendering. 2. The introduction of practical strategies such as early termination and self pruning also make sense to me. 3. The proposed method can reconstruct photorealistic images by learning high-resolution volumes. The comparison against previous methods show that the proposed method generate results of higher quality.

Weaknesses: 1. One obvious shortcoming of the proposed method compared to NeRF is that it will require a much larger storage. I would hope that the authors add a comparison on storage required and discuss it in the paper. 2. It is confusing to me that the performance of NeRF is much worse than what the original authors show in their paper. For example, in the Caterpillar scene, the result of NeRF is very blurry. However, I don't think such a scene is more complex than those used in NeRF (such as the Lego scene). Similarly, in the Steamtrain scene, the result of NeRF has some floating artifacts, which is not normal for synthetic scenes with a clean background. I am wondering whether the authors have made a fair comparison to NeRF. 3. Are there any failure cases where the network prune out voxels incorrectly? How robust is the self pruning scheme? In Line 176, the authors say in most of the experiments the threshold is set to 0.5, so are there any insights on how such a parameter should be adjusted for different scenes? 4. In Figure 3, when the ratio of the foreground is increasing, the rendering time is not monotonically increasing. What's the reason for it?

Correctness: The method is technically correct.

Clarity: The paper is well written.

Relation to Prior Work: The related work is good.

Reproducibility: Yes

Additional Feedback: Overall I like the idea of applying sparse architectures for volumetric representations. Such an efficient representation could benefit many other tasks in CV and CG. Post-rebuttal: The rebuttal addresses my concerns. The authors should update the baseline results in the final version.

[Author Response · NeurIPS 2020]

We thank reviewers for the constructive comments. We will release code and data. We now address main concerns.

**Synthetic results are better than real scene results (R2, R3, R4):** The camera pose errors of real scene data, albeit
small, caused this difference, since there are no camera pose errors for synthetic data.

**Storage usage for network weights (R1, R2, R4):** The storage usage for the network weights of NSVF varies from
$3.2 \sim 16$MB (including around 2MB for MLPs), depending on the number of used voxels ($10 \sim 100$K). NeRF has two
(coarse and fine) slightly deeper MLPs with a total storage usage around 5MB. We will add this to the revision.

**Scene edtiting is not clear regarding MLPs (R1):** We used the learned multi-object model which is trained with
different voxel embeddings for each object but sharing the same MLPs (L224-225). We will make it clear.

**Training with RGB or RGBD images? (R2):** All the scenes except the ScanNet scenes are recovered from RGB
images. Our method is also applicable to RGBD data, e.g. ScanNet, for which depth is used for voxel initialization and
training. Fig. 16 shows a comparison of w/ and w/o voxel initialization. We will make the type of training data clear.

**How voxel size affects rendering speed (R2):** Large voxels used to bound a scene are likely to contain more empty
space, thus leading to longer rendering time spent on evaluations in empty space.

**Performance with different voxel resolutions (R3):** The ablations of different voxel resolutions w/ and w/o fixing
step size at different training rounds are shown in Table 2 and Table 4, respectively.

**Comparion with one round of training at the final resolution (R3):** Our test on *Wineholder* shows that compared
with one-round training, our progressive training is faster and easier to train, uses less space and achieves better quality.
The metrics (PSNR↑, SSIM↑, LPIPS↓) are: 29.77, 0.946, 0.033 (One-round) v.s. 32.04, 0.965, 0.020 (Progressive).

**Experiments on large-scale scenes and dynamic scenes (R3):** Table 1 (below) shows that NSVF achieves the best
performance on these two tasks. For the *ScanNet* results, better represented geometry results in better rendering quality.

Table 1: Results for *Maria Sequence* (Left) and *ScanNet (one scene)* (Right). Here geometry accuracy is measured by RMSE of ground-truth depths and depths of rendered geometry. No result for NV is reported for *ScanNet* because training failed to converge.

| | PSNR↑ | SSIM↑ | LPIPS↓ | | RMSE↓ | PSNR↑ | SSIM↑ | LPIPS↓ |
|---|---|---|---|---|---|---|---|---|
| SRN | 29.12 | 0.969 | 0.036 | SRN | 14.764 | 18.25 | 0.592 | 0.586 |
| NV | 33.86 | 0.979 | 0.027 | NeRF | 0.681 | 22.99 | 0.620 | 0.369 |
| NeRF | 34.19 | 0.980 | 0.026 | Ours (w/o depth) | 0.210 | 25.07 | 0.668 | 0.315 |
| Ours | **38.92** | **0.991** | **0.010** | Ours (w/ depth) | **0.079** | **25.48** | **0.688** | **0.301** |

**Effect of early termination (R3):** The quantitative metrics (PSNR↑, SSIM↑, LPIPS↓) and average rendering speed
(sec/frame) on *Wineholder* of NSVF with early termination with different thresholds are shown as follows: 31.93, 0.965,
0.021, 4.0 ($\epsilon = 0.0$) v.s. 32.03, 0.965, 0.020, 2.1 ($\epsilon = 0.001$) v.s. 32.04, 0.965, 0.020, 2.0 ($\epsilon = 0.01$) v.s. 29.99, 0.947,
0.029, 1.7 ($\epsilon = 0.1$). The selection of $\epsilon = 0.01$ gives the best trade-off between quality and rendering speed.

**How the initial grid resolution affects the performance (R3):** Our tests show that the initial grid resolution does not
affect the quality of results. We will include the experiments in the revision.

**Comparison with DeepVoxels (R3):** As stated in the SRN paper, SRN outperforms DeepVoxels (by the same authors).
So, as treated in NeRF, we see no need to compare with DeepVoxels because our method outperforms SRN.

**Eq. 2 seems incorrect (summation v.s. product) (R3):** Eq. 2 is correct, because it is equivalent to the one in
Mildenhall et al., based on the elementary identity $\exp(\sum_i x_i) = \prod_i \exp(x_i)$.

**Describe the benefits of sparse voxel grids (R3):** The benefits are described in detail throughout the paper, e.g., Line
48-51 on the benefits, Sec. 2 and Sec. 3 on the motivation and advantages of sparse voxel grids, etc.

**Why NeRF is better in SSIM (R3):** In fact, NeRF has worse SSIM scores than ours. In the submission we cited the
SSIM scores reported in the NeRF paper for the eight NeRF's synthetic objects. After submission, we were informed by
the NeRF's authors that their SSIM metric was calculated incorrectly. Now the corrected SSIM scores reported in their
updated version are $\sim 0.03$ lower than the original wrong scores. Thus, our method is better than NeRF in SSIM now.

**NeRF results for *Steamtrain* (R4):** Thanks for pointing this out. We realized after submission that we forgot the
preprocessing step of scaling the two models, *Steamtrain* and *Ignatius*, into the cube of side length 2 centered as
required for running NeRF code. We subsequently retrained NeRF for these two models with this preprocessing. The
results have improved but still are worse than our results. The corrected quantitative metrics (PSNR↑, SSIM↑, LPIPS↓)
for these two models are as follows: *Steamtrain*: 30.84, 0.966, 0.031 (NeRF) v.s. 35.13, 0.986, 0.010 (Ours); *Ignatius*:
25.43, 0.920, 0.111 (NeRF) v.s. 27.91, 0.930, 0.106 (Ours). We will correct the results of these two models in revision.

**The threshold for self pruning (R4):** We clarify that for ALL the experiments we set the threshold as 0.5, which
works stably. Self pruning may prune incorrectly for very thin structures. We will discuss failure cases in the revision.

**Other clarifications:** (i) the ratio of foreground to background is based on image pixel and it can reach 1 (R2); (ii)
we train all the methods with the same views (R3); (iii) "overfitting" refers to overfitting to training views (R3); (iv)
rendering time is related to not only the foreground ratio but also the complexity of the object itself (R4).

**We will also:** (i) add missing references (R2, R3); (ii) rephrase the statement "the proposed method is 10 times faster
than the state-of-the-art" (R3); (iii) summarize the experimental settings described in Appendix in the main paper (R3);
(iv) improve grammar and word usage, fix typos, and rewrite unclear parts (R3).

[Meta-Review · NeurIPS 2020]

All four expert reviewers were in favor of acceptance and the AC is inclined to agree. The AC would encourage the authors to incorporate all of the detailed results from the rebuttal, not only eg the corrections to the numbers but also all of the clarifications.